# Investigating Bio-Inspired Degradation of Toxic Dyes Using Potential Multi-Enzyme Producing Extremophiles

**DOI:** 10.3390/microorganisms11051273

**Published:** 2023-05-12

**Authors:** Van Hong Thi Pham, Jaisoo Kim, Soonwoong Chang, Donggyu Bang

**Affiliations:** 1Department of Environmental Energy Engineering, College of Creative Engineering of Kyonggi University, Suwon 16227, Republic of Korea; 2Department of Life Science, College of Natural Science of Kyonggi University, Suwon 16227, Republic of Korea; jkimtamu@kyonggi.ac.kr; 3Department of Environmental Energy Engineering, Graduate School of Kyonggi University, Suwon 16227, Republic of Korea; ahr1emd@kyonggi.ac.kr

**Keywords:** bacterial dye degradation, bioremediation, dye toxicity, extremophiles

## Abstract

Biological treatment methods overcome many of the drawbacks of physicochemical strategies and play a significant role in removing dye contamination for environmental sustainability. Numerous microorganisms have been investigated as promising dye-degrading candidates because of their high metabolic potential. However, few can be applied on a large scale because of the extremely harsh conditions in effluents polluted with multiple dyes, such as alkaline pH, high salinity/heavy metals/dye concentration, high temperature, and oxidative stress. Therefore, extremophilic microorganisms offer enormous opportunities for practical biodegradation processes as they are naturally adapted to multi-stress conditions due to the special structure of their cell wall, capsule, S-layer proteins, extracellular polymer substances (EPS), and siderophores structural and functional properties such as poly-enzymes produced. This review provides scientific information for a broader understanding of general dyes, their toxicity, and their harmful effects. The advantages and disadvantages of physicochemical methods are also highlighted and compared to those of microbial strategies. New techniques and methodologies used in recent studies are briefly summarized and discussed. In particular, this study addresses the key adaptation mechanisms, whole-cell, enzymatic degradation, and non-enzymatic pathways in aerobic, anaerobic, and combination conditions of extremophiles in dye degradation and decolorization. Furthermore, they have special metabolic pathways and protein frameworks that contribute significantly to the complete mineralization and decolorization of the dye when all functions are turned on. The high potential efficiency of microbial degradation by unculturable and multi-enzyme-producing extremophiles remains a question that needs to be answered in practical research.

## 1. Introduction

Comprising two-thirds of the more than 1000 available dyes, textile dyes are pollutants with harmful impacts on the environment, including air and water ecosystems which are already heavily loaded with inorganic/organic matter, pathogenic microorganisms, and toxic chemicals, including recalcitrant dyes, heavy metals, sulfides, and detergents. These factors negatively impact soil productivity, soil microbial communities, seed germination, and plant growth, ultimately affecting global carbon cycling [1,2]. Dyes cause aesthetic damage to water bodies and reduce the rate of photosynthesis in aquatic plants by inhibiting the penetration of light into water. Therefore, clean-up strategies are required for physicochemical treatments, bioremediation, and their combinations. However, dyes are difficult to remove from water using conventional methods due to their high solubility and the presence of multiple complex pollutants [3,4]. Additionally, these treatments result in incomplete mineralization and conversion to CO_2_ production from intensive sludge as a consequence of disposal. Further, the high cost of these treatments limits their application, leading to increased ecotoxicity in the long term [5,6]. Hence, biodegradation has been evaluated as an efficient alternative approach that has attracted research attention because it overcomes the drawbacks of traditional techniques and is economical, eco-friendly, and sustainable.

Recently, potential bacterial candidates were reported to play a crucial role in the biological degradation of dyes through the action of either extra/intracellular enzymes or the whole bacterial cell [7,8,9]. In particular, extremophilic and multi-enzyme-producing bacteria have received much attention, focusing on their metabolic pathways, with extraordinary properties being applied to bioremediation strategies, including dye decolorization. They can survive under stress and degrade synthetic dyes into non-colored compounds or mineralize them partially or completely under certain environmental conditions. Dye effluents with extremely variable compositions and intermediate products generated during dye degradation may induce oxidative stress in microorganisms. Therefore, only bacterial strains have shown the ability to withstand harsh conditions and produce multiple enzymes to protect themselves from toxic environments which may enhance the efficiency of dye degradation compared to non-specific microorganisms.

Moreover, industrially-produced azo dyes are xenobiotic compounds that are considered recalcitrant to biological degradation processes [10]. Le Borgne et al. suggested mechanisms of xenobiotic degradation by halophilic bacteria involving the production of microbial enzymes and catabolic genes responsible for toxicant degradation [11]

Additives used in the textile industry lead to the formation of refractory intermediates which are toxic pollutants such as polycyclic aromatic hydrocarbons (PAHs) and aromatic amines [12,13]. Under highly saline conditions in dye effluents, the degradation of PAH occurs slowly. Thus, halophilic bacteria have been proposed to have high potential because of the valuable amount of biosurfactants that can enhance the degradation of PAH [14].

In another study, Pham et al. introduced a facultative bacterial strain that can grow and degrade Methylene Blue at an acidic pH of 5 through fundamental mechanisms of bioabsorption using both dead and living bacterial cells and accumulation [9]. The biodegradation of dyes occurs effectively under both aerobic and anaerobic conditions, or in combination. Thus, aerobes, anaerobes, and facultative bacteria play significant roles in degradation efficiency.

Some dye particles exhibit a high chemical structural class and are exclusively degraded by a few enzymes such as ligninolytic enzymes, which are recognized as efficient tools for dye degradation because of their extracellular and nonspecific nature [15,16]. In addition, azoreductases and laccases have shown great potential for decolorizing a wide range of industrial dyes [7,17]. However, bacterial dye degradation has proven to be a major challenge in wastewater under the diverse stresses of high salt concentrations, various contaminated metals, and the complex nature of waste [18]. Recently, halophilic bacteria have been investigated as potential dye degraders [18,19,20]. Surprisingly, these candidates can survive harsh temperatures, pH, and heavy metal conditions [9,21]. Other thermophilic microorganisms also exhibit adaptive responses to oxidative stress during dye degradation [22,23]. Nevertheless, these studies provide insights into the enzyme-producing potential of these extremophilic bacteria are limited so far. Thus, this review put an effort to mainly focus on analyzing the potential of extremophilic bacteria, the mechanisms of their adaptation to the harsh conditions of polluted environments, and their metabolism involved in the decolorization of dyes via whole-cell and secreted multi-enzyme pathways as well as the importance of developing methods for isolating these promising bacteria to be used as green materials for dye biodegradation in the future.

## 2. Toxicity and the Negative Impact of Dyes on Human and Environmental Ecosystems

The long-term release of untreated dye-containing wastewater can cause serious ecotoxicological threats to multiple environmental ecosystems, especially aquatic life, soil fertility, and crop germination rates in the life cycle [24,25].

The presence of color associated with dyes increases the chemical oxygen demand (COD), biological oxygen demand (BOD), total organic carbon (TOC), and suspended solids which reduce photosynthesis, promote toxicity of several intermediate amino acids, inhibit plant growth, and create recalcitrance and bioaccumulation [26,27,28]. Furthermore, the residuals of chemicals used during dye manufacture can evaporate and pollute the air. Long-term or accidental exposure to these residuals may cause respiratory problems after inhalation or trigger allergic reactions causing skin irritation and itching after absorption through the skin. Water-soluble azo dyes are extremely toxic when metabolized by live enzymes.

Moreover, dyes ingested and accumulated in fish can become toxic intermediates for both fish and their predators including humans [29]. Traditional techniques and limited biodegradability lead to recalcitrance and bioaccumulation and may promote toxicity, mutagenicity, and carcinogenicity [6,30].

Dyes contain heavy metals, such as Lead (Pb), Arsenic (As), Chromium (Cr), Nickel (Ni), Copper (Cu), Cadmium (Cd), Mercury (Hg), and Zinc (Zn) which act as catalysts, oxidizing agents, fixing agents, and cross-linking agents in the dyeing process. They may accumulate in the food chain causing harmful effects on human health and considerably damage the growth and development of plants [31]. Oxidative stress caused by Cr in textile dyes is another problem associated with the recalcitrant characteristics that affect photosynthesis and CO_2_ assimilation [32].

Some of the negative effects of dyes on humans and environmental ecosystems are shown in Figure 1.

## 3. The Role of Polyextremophilic Bacteria in Dye Degradation

Extremophiles are classified into different categories based on the harsh environments to which they have adapted. However, many extremophilic bacteria belong to two or more categories and are known as polyextremophiles [33]. Some studies have highlighted the benefits of extremophilic and polyextremophilic microorganisms in biotechnology and bioremediation. Synthetic dyes often exist as complex mixtures of structurally different types, in addition to other contaminants such as detergents, surfactants, heavy metals, and high concentrations of salts caused by sodium hydroxide in dye baths under diverse pH and temperature stresses [34]. Additionally, high concentrations of lead salts may inhibit decolorization because survival and activity decrease under extreme conditions. Thus, there are obstacles to the biological treatment of dye-containing effluents, particularly for large-scale applications. Biological approaches using polyenzyme-producing extremophiles have shown great efficiency in the degradation of aerobic and anaerobic dyes [35]. Therefore, investigating well-adapted bacterial candidates under multiple stress conditions is a promising strategy for research and application for removing dyes from wastewater.

The general mechanism of dye removal by bacteria involves the binding of dye molecules to specific groups on the surface of bacterial cells, such as alcohol, aldehydes, carboxylic, ether, and phenolic groups (biosorption) and their gradual accumulation inside the cell (bioaccumulation) [36]. Biosorption is carried out in living or dead cells, whereas bioaccumulation involves the active uptake and accumulation of dye molecules inside living cells. However, in the case of dead cells as an effective biosorbent, the negative effects of its disposal need further study.

During the process of microbial removal of dyes, reactive oxygen species (ROS), such as superoxide anion radicals (^•^O_2_^−^) and hydroxyl radicals (^•^OH), are generated during the dye degradation process due to the contribution of high oxygen pressure, lack of water, high temperature, high metal ion concentrations, radiation, high salinity, and other chemicals that cause cells to face oxidative stress (OS). Therefore, protective mechanisms of extremophiles against damage by enzyme systems, including superoxide dismutase (SOD), glutathione peroxidase (GPx), and catalase (CAT), play a role in protecting the cell from OS generated during the dye degradation process [37]. Moreover, extremophiles have special protein adaptations to each environmental stress condition, including genetic changes resulting from changes in protein sequences and structures. For example, thermophilic bacteria are exposed to oxidative stress, which directly or indirectly damages various metalloproteins. They are active in effective DNA repair systems, antioxidant defense systems, selective protection against oxidative protein damage, and removal of damaged macromolecules. On the other hand, thermophiles have different amino acid contents than ordinary proteins to increase thermal stability with the number of large hydrophobic residues or cores, a greater number of disulfide bonds/bridges, and increased interactions of ions. In addition, thermostable thermophilic proteins contain high amounts of arginine which leads to an increase in salt bridge formation to stabilize thermophilic proteins [38].

Different physicochemical methods have been used for dye removal from textile industry wastewater; however, they are costly, non-environmentally friendly, and cause immediate secondary toxicity. To overcome several physicochemical and environmental constraints, such as high water viscosity, osmotic stress, and high gas solubility, psychrophiles modify the fatty acid composition of the cell membrane by notably increasing the proportion of unsaturated fatty acids and modulating the activity of enzymes involved in lipid biosynthesis [39]. With high glycine content in the proteome, psychrophiles have greater conformational mobility and stability at low temperatures because of the large number of disulfide bridges in their proteins. Additionally, psychrophiles have developed various adaptive mechanisms at the molecular level. Psychrophilic enzymes have large cavities that improve their flexibility and produce anti-ice-binding proteins that prevent the formation of ice crystals inside the cell [38,40].

Faced with the other detrimental effects of low water activity and salt interference, low hydrophobicity is considered an adaptation mechanism of halophiles at the protein level that allows them to exhibit an affinity toward toxicants. Moreover, increasing the negative surface charge on proteins helps them become more soluble and flexible at high salt concentrations in dye effluents. Dinucleotides are abundant in the genomes of halophiles and have specific genomic signatures for hypersaline adaptation that are significantly different from those of non-halophiles [41]. Azoreduction by halophilic bacteria has been effectively performed as a nonspecific reductive process mediated by an enzyme system [19].

Under highly acidic conditions, acidophiles have adaptive mechanisms, such as building a waterproof cell membrane and cytoplasmic buffering to secrete acid and maintain natural pH conditions by transporting protons across the cell membrane [42,43]. Further reactions to dye breakdown involve a specific mechanism involving different metabolic enzymes.

In an alkaline environment, alkaliphiles have the most significant effects on cell-free protein synthesis systems to adapt. To maintain a neutral intracellular pH, the internal pHi of alkaliphiles can be estimated from the optimal pH of the intracellular enzymes. For example, α-galactosidase from alkaliphilic *Micrococcus* sp. strain 31-2 and β-galactosidase from *Bacillus circulans* sp. *alkalophilus* have an optimal catalytic pH of 7.5, and pH 6 to 7.5, respectively [44,45]. Moreover, multi-cation/proton antiporters (CPAs) such as Na^+^, K^+^, Ca ^2+^, and H^+^ in the environment enhance the electrical membrane potential to provide the energy for ATP synthesis that is adequate to support cell growth under extreme pH conditions. The structure of cell wall polymers, which mainly consists of teichuronic acids and teichuronopeptides, also greatly contributes to maintaining pH homeostasis in the cytoplasm via a negatively charged matrix and reduces the pH value at the cell surface of alkaliphilic bacteria [46]. Alkaliphilic bacteria in pure cultures or consortia, such as *Bacteroides* spp., *Eubacterium* spp., *Clostridium* spp., *Proteus vulgaris*, *Streptococcus faecalis*, *Bacillus* spp., and *Sphingomonas* are effective azo dye-degrading candidates [38].

The adaptation mechanisms and dye degradation of dye of the main functional extremophiles under severe conditions was summarized in Figure 2. 

In recent studies, some thermophiles have been investigated as highly adaptive response candidates for redox stress during the dye degradation process of textile wastewater [36]. A halophilic and alkalophilic bacterial strain of *Bacillus albus* DD1 was identified as an azo dye RB5 degrader with 98% efficiency [47]. Another member of the genus *Bacillus* isolated from an alkaline lake showed complete dye degradation efficiency under anoxic and anaerobic conditions at pH 10 [48]. *Acinetobacter baumannii* strain was investigated as a degrader of Reactive Red 198, with a removal efficiency of 96.2% under hypersaline conditions [49]. A previous study illustrated that thermophilic anaerobic treatment is an interesting approach for enhancing the decolorization of azo dyes with a significant contribution from thermophiles [50]. In another study, the dye Remazol Brilliant Blue R (RBBR) was determined to have a 90% removal efficiency by immobilized cells of the thermophilic bacterial strain *Geobacillus stearothermophilus* ATCC 10,149 [51]. However, there are few reports published on azo dye removal by psychrotolerant bacteria so far [52]. Interestingly, the psychrophilic bacterial strain *Psychrobacter almentarius,* isolated from seawater sediment, was able to decolorize three reactive dyes: Reactive Black 5, Reactive Golden Ovifix, and Reactive Blue BRS [53]. Recently, a psychrotolerant, halophilic, alkalophilic, and xenobiotic degrader, Actinobacterium *Zhihengliuella* sp. ISTPL4 has been explored for its potential to degrade the azo dye methyl red [54,55].

In another study, the four bacterial strains, namely *Shewanella indica* strain ST2, *Oceanimonassmirnovii* strain ST3, *Enterococcus faecalis* strain ST5, and *Clostridiumbufermentans* strain ST12 demonstrated high tolerance to salinity (>20 g L^−1^), temperature (35–50 °C), and pH (<4 and >8), as well as the presence of metals except for Cd, Cu, and Pb [56]. Halotolerant, alkali-thermo-tolerant bacterial mixed cultures have been investigated as cost-effective biodegraders of Direct Red 81 (DR81) with ≥70% decolorization within 24 h of dye up to 600 mg/L at 60 °C, pH 10, and 5% salinity [57]. Many thermophilic bacteria have shown the potential for treating textile industry effluents with high salinity [58]. Typical dye-degrading bacterial strains are listed in Table 1:

## 4. Enzyme-Linked Bioremediation of Dyes

In recent decades, the enzymatic processes carried out by “biocatalysts” for the treatment of dye-contaminated wastewater have been considered because they overcome the drawbacks of physicochemical methods. The currently used dye removal methods are summarized in Figure 3. Several enzymes, such as hydrolases, laccases, azoreductases, and lignin peroxidases, are effective in cleaving the aromatic rings and amines of dye molecules [9,48,81,82]. Specifically, the azoreductase activity of a mixed extremophilic bacteria was found to be optimal at 70 °C and stable at temperatures above 50 °C and in a wide pH range of 4–9 [57]. Bacterial degradation of dyes can be carried out in the presence of various substrates. For example, Navitan Fast Blue S5R is aerobically degraded by *Pseudomonas aeruginosa* in the presence of glucose as a carbon source for bacterial growth and metabolism [83]. In another study, *Aeromonas hydrophila* was shown to degrade Crystal Violet dye using laccase enzymes and lignin peroxidase with sucrose and yeast extract as substrates [84]. Pham et al. found that *Bacillus* sp. React3 is capable of degrading up to 97% of Methylene Blue using lignin peroxidase in the presence of tryptone and yeast extract [9].

The common mechanisms of biological degradation of dye effluents using bacterial enzymes usually involve two steps. First, NADH or NADPH produced by azoreductases from bacteria serve as electron donors for the cleavage of the azo linkage (N=N) and form toxic metabolites or aromatic amines. Second, the toxic metabolites further transform into non- or less toxic compounds [57]. Azoreductases are enzymes that are more active under anaerobic conditions than during aerobic processes or respiration [85]. However, some previous studies have illustrated that these azoreductases behave differently with no clear degradation patterns [86,87]. The real physiological role of azoreductases needs to be studied more to answer correctly the still questionable activity of azo dye consumption [87].

Laccases are multicopper oxidase enzymes that are important biocatalysts for various industrial applications. These enzymes lack cofactors but use molecular oxygen as an electron acceptor to carry out their oxidation reactions, making them more versatile and able to use various substrates by adding redox mediators to the reactions. They can degrade dyes through non-specific free radical-mediated mechanisms without yielding hazardous products. Moreover, they are active over a wide pH range, stable at high temperatures, and tolerant to detergents [88]. Therefore, microbial laccases have potential opportunities for commercial applications. A recent study investigated laccases produced from *Bacillus cereus* (*B. Cereus*) and *Pseudomonas parafulva* (*P. parafulva*) and the enzyme activity was optimal at 50 °C [17]. In another study, laccases produced by *Bacillus* sp. NU2 showed a high catalytic potential with optimal activity at 60 °C, and pH 8 for the detoxification of various dyes such as Congo Red, Methyl Orange, Remazol Brilliant Blue R, Reactive Blue 4, and Malachite Green [89]. Laccases that served as good dye decolorizing agents have been produced from the thermophilic bacterial species *Streptomyces psammoticus* and *Stenotrophomonas maltophilia* [90,91].

Catalase was determined to the role of protecting bacteria against oxidative damages caused by dye [37,92]. Further research has identified a catalase and a laccase isolated from the extremophile *Geobacillus thermo pakistaniensis* which can bleach colors [93].

Hydrolase enzymes containing protease, amylase, and cellulase are responsible for breaking bonds of C-N; C-C; and C-O in dye degradation. Previous studies have investigated the contribution of amylases produced by *Bacillus megaterium* and *Anoxybacillus rupiensis* and a serine protease produced from *Bacillus* sp. were investigated in the previous studies that exposed capacity in the degradation of dyes [93,94,95]. The glucose oxidase enzyme can remove natural pigments through bio-bleaching by glucose oxidation mechanisms [96]. In the dye textile industry, stain and color preservatives can be removed from fabrics using the cellulase Puradax HA produced by *Bacillus* sp. [97]. Alkali-thermophilic thermozymes have been evaluated in bioscouring processes at alkaline pH and high temperatures [98]. Alkaline pectinases generated by *Bacillus* sp. and *Pseudomonas* sp. can be used to protect cellulose and fiber from damage [94,99]. Recently, cold-adapted amylases and proteases that can eliminate stain-containing starch were successfully commercialized [100]. In another study, alkaline protease produced from bacterial consortia has been shown the decolorization capacity of Direct Red-81 (68.8%) and Direct Organge-34 (70.78%) [101].

Peroxidases, including manganese peroxidases and lignin peroxidases, are being discovered as they are directly involved in the degradation of dyes and xenobiotics. In particular, bacterial peroxidases are preferred to fungal peroxidases because bacteria are more flexible during protein engineering to enhance catalyst activity [102]. *Thermosediminibacter* spp. have been investigated as promising candidates for the production of peroxidases (DyP), laccases, and azoreductases involved in dye decolorization [103]. Lignocellulosic biomass is currently considered not only an abundant renewable source for the production of biofuels and valuable bioproducts, but also a significant substrate or biosorbent for dye removal because of its eco-friendly nature and natural availability [9]. Lignin is one of three different polymers decomposed from lignocellulose and may be a suitable substrate for extremophilic lignin peroxidase-producing bacteria to enhance LiP production which contributes to the dye degradation process. The promising halotolerant and alkalophilic *Bacillus ligniniphilus* L1 has illustrated high-value utilization of lignin at pH 9 [27].

Compared to other extremozymes, psychrophilic enzymes have lower thermal stability, higher structural flexibility, and greater specific activity [104]. On the other hand, superoxide dismutases (SODs) are antioxidant enzymes that play important roles in the cellular defense against harmful environmental factors. Both SODs and catalase (CAT) enzymes are found in the thermophilic bacterium *Exiguobacterium profundum* [36].

Although extremozymes have been investigated for several decades, their characteristics, structures, and functions have changed over time. Therefore, further studies on the potential activities of these extremophiles and their enzymes in certain conditions to obtain maximum efficiency in biological applications are needed, and would be an exciting venture for researchers.

## 5. Discussion and Future Prospective

The utilization of microorganisms in the dye degradation process has increased due to their unique metabolic pathways and protein frameworks that help them mineralize and completely decolorize the dye under specific ecological conditions [105]. In-depth studies on the enzymes, genes, and metabolic mechanisms responsible for the decolorization by extremophiles are currently needed for application in wastewater treatment. The possibility of identifying, isolating, cloning, and transferring genes that encode the degrading enzymes should be explored to identify potential superdegrading microorganisms from extremophiles.

Several studies have demonstrated that bacterial biomass is a promising biosorbent material for textile dye bioremediation because it provides sources of carbon and nitrogen [9,106]. Therefore, hybrid adsorbent systems are highly efficient for dye removal at low operating costs. However, the mechanisms involved in the link between the living/dead cell biomass and dyes are complex. Moreover, the consequences of final biomass disposal and dye effluents after absorption are still uncontrolled and neglected.

Agrowaste is considered an excellent substrate source for polyenzyme production in the fermentation process, eliminating hemicellulose, cellulose, and lignin [9,107]. Therefore, multi-enzyme-producing bacteria are potential candidates for various waste decompositions in the polypolluted form. Interestingly, most bacterial strains which have high adaptation under stress have been investigated as multiple enzyme-producing candidates in recent studies [9,108]. Therefore, more practical research regarding the ecology, taxonomy, and molecular properties of these special bacteria should be carried out to better understand the correlation between the high extreme tolerance and enzyme-producing capacity during the degradation process of toxicants.

The biological degradation of dyes can be performed in the presence or absence of oxygen. Although anaerobic conditions are mostly responsible for the degradation of azo dyes in the first step of azo-linkage cleavage, the aromatic amines formed in the second step can be degraded by bacteria almost exclusively under aerobic conditions. Thus, the combination of both anaerobic and aerobic phases in one treatment process may speed up the process and improve its efficiency [109].

Other important biocompounds that should also be considered are extremolytes. These are small organic molecules that accumulate inside cells and are either synthesized or taken up by extremophilic bacteria. They play a crucial role in protecting the macromolecules and cell structures of extremophiles by forming and stabilizing protective water layers [110]. In addition, the biosurfactants secreted by extremophiles during dye degradation should be further studied in terms of changes in their composition and function under certain operating conditions.

Because extremophiles have huge potential applications with superior performance in a wide range of areas, they need to be considered as a great replacement for many current conventional methods. Developing novel isolation methods to investigate unculturable or fastidious bacteria such as designing the nutrient components of culture media, equipment, and optimizing the growth conditions based on characteristics, adaptation mechanisms, and metabolic pathways should be continued as an independent further step. However, studying these polyenzyme-producing extremophiles is more challenging because of the lack of knowledge about particular physiological features and molecular properties due to the change in their complex metabolic pathways under extreme environments. However, the long-term effects of using extremophilic bacteria and their by-products should be verified with each original dye effluent input.

## 6. Conclusions

The specificity of special biological molecules, complex metabolic pathways, and robustness enables extremophiles to easily face multiple environmental stresses. Thus, they have attracted significant attention as powerful sources for biological applications and industrial purposes and the development of bioremediation techniques using biomass and enzymes as super catalysts. The demand for extremophilic bacteria and multi-extremozymes and their potential applications have increased tremendously. Therefore, it is expected that new methods, both dependent and independent, will be developed to provide a pool of extremely competitive cells and catalysts for waste treatment and other bio-based applications. It is necessary to figure out the link between the capacity of enzyme production and the removal rate of dye under each certain extreme conditions based on the components of the original dye effluent that may affect the efficiency of the dye degradation process. Additionally, understanding the changes in the metabolic pathways and physiological properties of polyenzyme-producing bacteria under harsh fluctuating environmental conditions may help in the discovery of novel enzymes and their potential functions. Moreover, insight into the genetic diversity, metabolic engineering, and extremolytes of these potential extremophiles is needed to further study their outstanding characteristics such as structure and biochemical properties and significant values to achieve sustainable development goals in environmental and industrial fields.

## Figures and Tables

**Figure 1 microorganisms-11-01273-f001:**
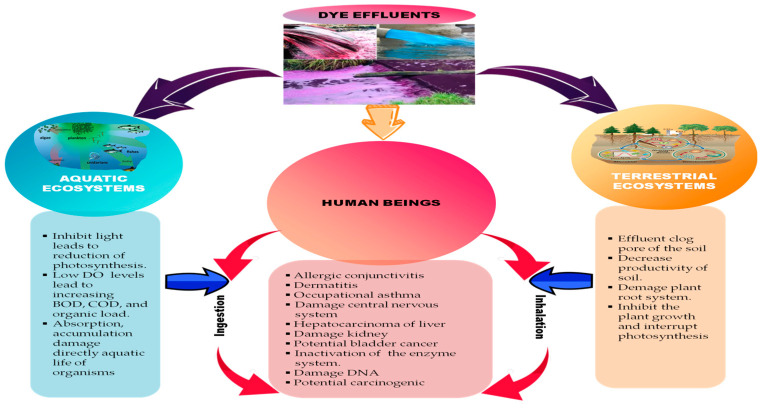
The hazardous effects of dyes on humans and the environment.

**Figure 2 microorganisms-11-01273-f002:**
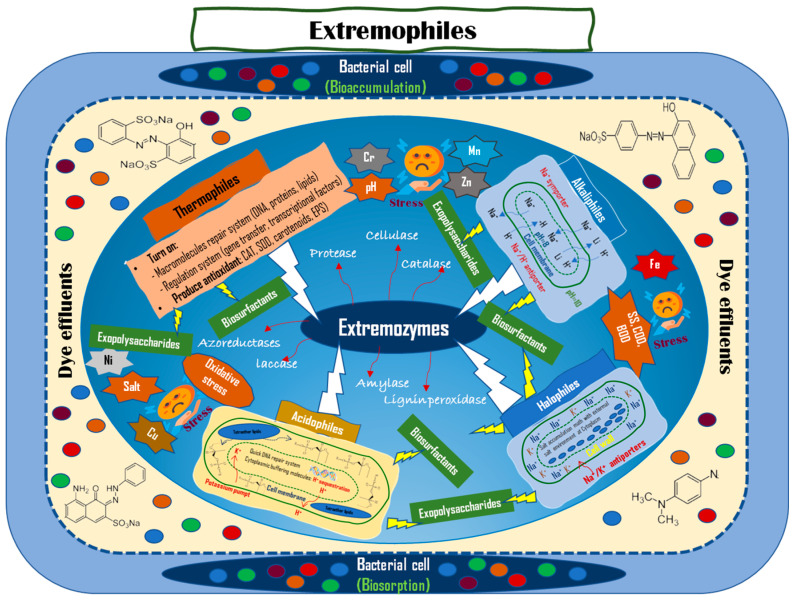
Mechanisms of adaptations and dye degradation of extremophiles under environmental stresses.

**Figure 3 microorganisms-11-01273-f003:**
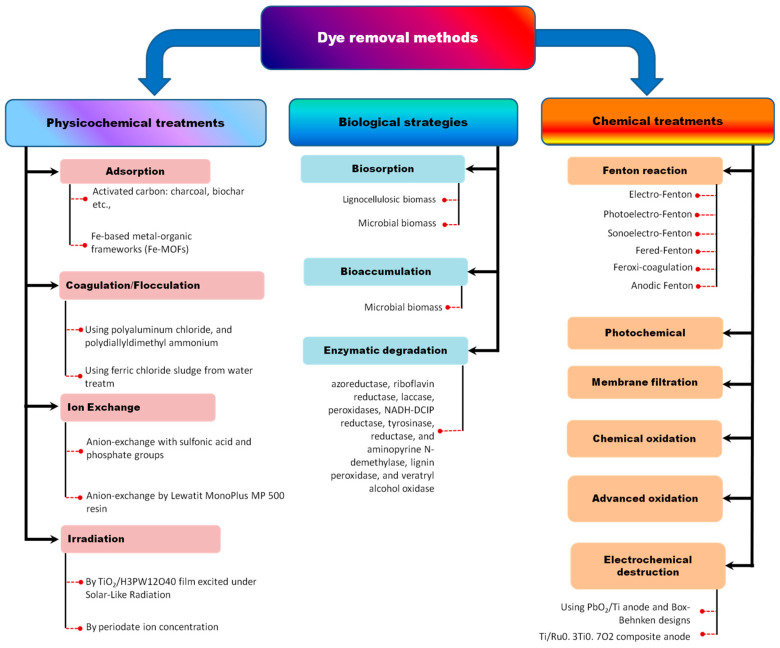
Current methods for dye removal.

**Table 1 microorganisms-11-01273-t001:** List of potential extremophiles and functional key enzymes/substrates in dye biodegradation.

Group	Name of Bacterial Strain	key Enzymes/Substrates	Type of Dyes	Efficiency	References
Thermophiles	*Geobacillus stearothermophilus* ATCC 10149	Extracellular laccase	Remazol Brilliant Blue R	90%	[59]
	*Anoxybacillus* sp. PDR2	Quinone oxidoreductase	Direct Black G	90%	[60]
	*Nivibacillus thermophiles* SG-1	Gene encoding riboflavin biosynthesis protein	Azo dye (Orange I)	100%	[61]
	Consortium of *Caldanaerobacter and Pseudomonas*	not reported	Acid Orange 7	90%	[62]
	*Caldanaerobacter*	Xylose	Acid Orange 7	97%	[63]
	*Geobacillus thermoleovorans* KNG 112	not reported	amaranth R.I and red E azo dyes	~98%	[64,65]
	*Geobacillus yumthangensis*	Laccase	Organic dyes, including Alizarin, Acid red 27, Congo red, bromophenol blue, Coomassie brilliant blue R-250, Malachite green, and Indigo carmine	~99%	[66]
	*Thermus* sp.	Laccase	xylidine	98%	[67]
	*Geobacillus thermoleovorans*	Not reported	Methylene Blue and Acid Orange G	100%	[68]
Psychrophiles	*Micrococcus antarcticus*	Psychrophilic β-glucosidase	Starch stain	not reported	[69]
	*Psychrobacter almentarius*	Not reported	Reactive Black 5, Reactive Golden Ovifix, and Reactive Blue BRS	~100%	[53]
	*Psychrobacter* sp.	Not reported	Fast orange, Methanil yellow, and Acid fast red	~85%	[70]
	*Zhihengliuella* sp.	Lignin peroxidase and laccase	Methyl red	98.87%	[55]
	*Bacillus* sp.	Azo-reductase	Amido black 10B, Evans blue, Janus green, Methyl orange, Methyl red, and orange G)	~98%	[52]
Alkaliphiles	*Bacillus licheniformis* LS04	Laccase	Reactive black 5	>80%	[71]
	*Bacillus fermus*	Not reported	Direct Blue-14	>92.76%	[72]
	*Nesterenkonia lacusekhoensis* EMLA3	Not reported	Methyl red	97%	[73]
	*Pseudomonas mendocina*	Laccase	Mixture of reactive red (RR), Reactive brown (RB), and Reactive black (RBL)	58.40%	[35]
	*Bacillus subtilis*	Intracellular azoreductase	Mixed azo dyes	87.35%	[74]
	*Halopiger aswanensis*	Lignin Peroxidase (LiP), Manganese Peroxidase (MnP), and laccase	Malachite Green, Methyl orange	~93%	[75]
Halophiles	*Halomonas* sp	Azoreductase	Acid Brilliant Scarlet GR		[76]
	*Halomonas* sp. strain A55	Not reported	different dyes	~100%	[77]
	*Nesterenkonia lacusekhoensis* EMLA3	Not reported	methyl red	97%	[73]
	*Mixture of Enterococcus*, unclassified *Enterobacteriaceae*, *Staphylococcus*, *Bacillus*, and *Kosakonia*.	Laccase, lignin peroxidase, manganese peroxidase, azo reductase, and NADH–DCIP reductase	Congo red, Direct Black G (DBG), Amaranth, Methyl red, and Methyl orange	~100%	[78]
	*Bacillus circulans* BWL1061	Azoreductase, NADH-DCIP reductase, and laccase	Methyl orange	>90%	[79]
	*Alcaligenes faecalis*	Azoreductase, laccase and NADH-DCIP (nicotinamide adenine dinucleotide-dichlorophenol indophenols) reductase	Acid Scarlet 3R	~100%	[80]
	*Bacillus* sp. strain CH12	Not reported	Reactive Red 239	~100%	[48]

## Data Availability

MDPI Research Data Policies.

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
