# Peer review of "Investigating Bio-Inspired Degradation of Toxic Dyes Using Potential Multi-Enzyme Producing Extremophiles"

_microorganisms, 2023, doi:10.3390/microorganisms11051273_

Round 1

Reviewer 1 Report

The authors performed detailed work regarding the topic. They studied bio-inspired degradation of toxic dyes using potential polyextremophilic and multi-enzyme-producing bacteria. The study summarizes various reports on the removal of toxic dyes. This review study is impressive. However, the authors need to communicate the novelty of this study, especially in the introduction part. What makes this manuscript significant compared to other similar reviews on this topic? What about the mechanism of degradation of toxic dyes, what are the major reactive oxygen species (ROS)? The authors should also include an image showcasing the mechanism of toxic dye degradation by polyextremophilic and multi-enzyme-producing bacteria. Is there no disadvantage to using these kinds of enzymes? Moreover, some of the references have a repetition of the numbers such as reference 1, 11 – 14, 32 – 35, 42, 51 – 76, 91 – 99, and 104.  

Author Response

Dear Reviewer,

Thank you very much for your valuable time and very helpful comments on our manuscript and giving us an oppportunity to revise it.

We have responsed your comments in the attached file and we also revised it based on your sugguestion in the text of manuscript.

Please kindly to check all of them. Thank you very much for all your help.

Best regards,

Pham, Van H.T

Reviewer 2 Report

The review “Studying on bio-inspired degradation of toxic dyes using  potential polyextremophilic and multi-enzyme producing bacteria” is presented well, but there are some serious deficiencies that need to be addressed properly.

1)      The abstract is somewhat not related to the main idea, it should be added with a clear presentation about the polyextremophilic and multi-enzyme producing bacteria.

2)      Section 2 is quite long in such case; it should be shortened so that the main theme of the review should not be distracted. Is it ok to discuss the toxicity and harmful effects of dyes but it’s a usual thing.

3)      Line 179-186, the main idea should be clearly presented, so that readers should understand it easily.

4)      The discussion and prospective portion, if possible, should be discussion and future prospectives, and also should be added with enough information.

5)      Lastly, the review lacks the required amount of knowledge about the main theme. Therefore, it is necessary to add more information related to the main idea without diverting from it.

Author Response

Dear Reviewer,

Thank you very much for your kind consideration with valuable time and very helpful comments on our manuscript and giving us an oppportunity to revise it.

We have responsed your comments in the attached file and we also revised it based on your sugguestion in the text of manuscript.

Please kindly to check all of them. Thank you very much for all your help.

Best regards,

Pham, Van H.T

Reviewer 3 Report

The manuscript shows a bibliographic review on the biodegradation of dyes using enzymes or biomass from bacteria. It is of potential interest to the readers of the Journal, an adequate structure of the manuscript is observed and its content is sufficient. It is suggested that the article can be published, only small typographical details should be corrected, for example:

Line 94, water-soluble instead of water -soluble

Line 196, azo instead of azdo

Line 208, … decolorize e three…, is “e” correct?

Lines 214, 215, separate the species from the genus “Oceanimonassmirnovii” and “Clostriumbufermentans

Line 219, separate (DR81)with

In Table 1, the first enzyme, correct word and letter format: Excellular laccase

Line 240, Pham et al. and co-workers, use only et al. or co-workers

Line 322. Missing point: cost However

Check that the format of all references is correct. There are many references with double number in the list

Author Response

Dear Reviewer,

Thank you very much for your valuable time and useful comments on our manuscript and giving us an oppportunity to revise it.

We have responsed your comments in the attached file and we also revised it based on your sugguestion in the text of manuscript.

Please kindly to check all of them. Thank you very much for all your help.

Best regards,

Pham, Van H.T

Round 2

Reviewer 1 Report

The authors have revised the manuscript as suggested.

Reviewer 2 Report

The authors have partially revised the manuscripts to some degree.